# Comorbidities of COVID-19 Patients

**DOI:** 10.3390/medicina59081393

**Published:** 2023-07-29

**Authors:** Radu Silaghi-Dumitrescu, Iulia Patrascu, Maria Lehene, Iulia Bercea

**Affiliations:** 1Faculty of Chemistry and Chemical Engineering, Babes-Bolyai University, 11 Arany Janos Str., 400028 Cluj-Napoca, Romaniamaria.lehene@ubbcluj.ro (M.L.);; 2Bistrita County Emergency Clinical Hospital, 42 General Grigore Bălan, Bld., 420094 Bistrita, Romania

**Keywords:** COVID-19, SARS-CoV-2, comorbidities, long-term effects

## Abstract

The novel Severe Acute Respiratory Syndrome Coronavirus 2 (SARS-CoV-2) responsible for the coronavirus disease outbreak initiated in 2019 (COVID-19) has been shown to affect the health of infected patients in a manner at times dependent on pre-existing comorbidities. Reported here is an overview of the correlation between comorbidities and the exacerbation of the disease in patients with COVID-19, which may lead to poor clinical outcomes or mortality. General medical issues are also reviewed, such as the types of symptoms present in people infected with SARS-CoV-2, the long-term effects of COVID-19 disease, and the types of treatment that are currently used.

## 1. Introduction

The severe acute respiratory syndrome coronavirus 2 (SARS-CoV-2), responsible for the Coronavirus disease 2019 (COVID-19) [1], is a member of the Coronaviridae family, with a 29 kb single-stranded RNA genome [2]. It employs its structural spike (S) glycoprotein to attach to the ACE-2 (angiotensin-converting enzyme 2) receptor protein on the surface of the host cell [3]. The S protein is composed of two subunits, S1 and S2. The S1 subunit is responsible for interaction with ACE-2, while the S2 subunit is involved in fusion with the cell. The very high affinity of protein S for ACE-2 is largely responsible for the increased infectivity of SARS-CoV-2 compared to other related viruses, such as SARS-CoV [4].

As expected of an RNA virus, SARS-CoV-2 tends to develop mutations, which has led to a number of well-described variants, the most notable of which are Alpha (B.1.1.7), Beta (B.1.351), Gamma (P.1), Delta (B.1.617.2), and Omicron (B.1.1.529) [5]. The rate of mutations is 0.8–2.4 × 10^−3^ nucleotide substitutions per site per year, which is in the same range as influenza A but at least an order of magnitude lower than for other RNA viruses [6]. According to initial studies, the R_0_ (reproduction number) of the SARS-CoV-2 virus was 4.7–6.6. With the help of vaccines, it decreased to approximately 1.5. The Omicron variant is the most transmissible, with an R_0_ higher than 10 [7].

The present manuscript attempts to offer a synthetic view of the situation related to comorbidities associated with COVID-19. The basis of the review was searches in the PubMed database for articles featuring the terms “COVID-19” and “comorbidity”. As the number of these articles was very large, meta-analyses were retained as the main sources. Additional references were added where clarification was needed beyond the information in the meta-analyses, including a number of pertinent suggestions from the Reviewers of the manuscript.

## 2. Symptoms of COVID-19

The incubation period of SARS-CoV-2 varies depending on the strain. Following 142 studies where 8112 patients were included, the mean incubation period for all strains was ~7 days—going as low as 3.4 for the Omicron variant. In people over 65 years old, the mean incubation period was ~7 days, as opposed to ~9 for children; severe illnesses led to a reduction of ~0.7 days in these values [8].

COVID-19 symptoms can vary from very severe to asymptomatic (none at all). Statistically, in 80% of people, the virus causes mild symptoms, but this largely depends on the type of SARS-CoV-2 strain, as well as on age; specifically, symptoms have been estimated to be observable in only 50% of the cases in children. Mild symptoms include fever, headache, sore throat, cough, diarrhea, vomiting, loss of taste and smell, and muscle pain. More severe symptoms are seen in some patients—dyspnea, shortness of breath, or abnormal chest imaging. In some patients, the lower respiratory tract is affected moderately, with oxygen saturation (SpO2) ≥94%. Patients with SpO2 < 94 and respiratory rates > 30 breaths/min or lung infiltrates > 50% represent the 4th category. Critical illness is the most serious category and entails septic shock, respiratory failure, and/or multiple organ dysfunction [9,10,11].

SARS-CoV-2 is mainly transmitted via respiratory droplets. It can also be transmitted through contaminated surfaces when a person touches the eyes, the nose, or the mouth with contaminated hands [12]. Asymptomatic and presymptomatic people are infectious. One to three days before the appearance of symptoms, people infected with SARS-CoV-2 can transmit the virus; they are responsible for the transmission of the virus in a proportion of 40–50% [10].

More severe symptoms appear especially in the case of older patients or those with other medical conditions such as the following: diabetes, cancer, obesity, HIV infection, chronic kidney disease, pulmonary hypertension, cerebrovascular disease, chronic liver diseases (alcoholic liver disease, cirrhosis, autoimmune hepatitis, and non-alcoholic fatty liver disease), chronic lung diseases (chronic obstructive pulmonary disease, bronchiectasis, pulmonary embolism, and interstitial lung disease), disabilities (ADHD, Down syndrome, spinal cord injuries, cerebral palsy, and congenital disabilities), cystic fibrosis, heart conditions (heart failure, coronary artery disease, cardiomyopathies, etc.), transplants, mental health disorders (mood disorders, schizophrenia spectrum disorders, and depression), primary immunodeficiency diseases, pregnancy, tuberculosis, smoking, receiving corticosteroids or other immunosuppressive medication [9].

A study was carried out in which post-COVID-19 symptoms were listed at 1–180 and 90–180 days after the acute phase of the disease: anxiety/depression (22.82% proportion of symptoms—1–180 days after the acute phase, and 15.49%–90–180 days after the acute phase); abdominal symptoms (15.58% proportion of symptoms—1–180 days after the acute phase, and 8.29%—90–180 days after the acute phase); abnormal breathing (18.71% proportion of symptoms—1–180 days after the acute phase, and 7.94%—90–180 days after the acute phase); fatigue/malaise (12.82% proportion of symptoms—1–180 days after the acute phase, and 5.87%—90–180 days after the acute phase); chest/throat pain (12.60% proportion of symptoms—1–180 days after the acute phase, and 5.71%—90–180 days after the acute phase); headache (8.67% proportion of symptoms—1–180 days after the acute phase, and 4.63%—90–180 days after the acute phase); cognitive symptoms (7.88% proportion of symptoms—1–180 days after the acute phase, and 3.95%—90–180 days after the acute phase); myalgia (3.24% proportion of symptoms—1–180 days after the acute phase, and 1.54%—90–180 days after the acute phase) [12].

Regarding the neurological symptoms, there are three possible hypotheses: the direct involvement of the nervous system, a secondary mechanism to systemic diseases and lung damage, or the appearance of symptoms as a result of immune-mediated post-inflammatory complications [12].

One of the most common symptoms of SARS-CoV-2 infection is headache. Regarding the pathogenesis of this symptom in the case of COVID-19, quite a few things are known. According to ICDH-3 (third edition of the International Classification of Headache), an explanation that could be a first hypothesis is that the headache is attributed to a systemic viral infection with symptoms like fever, cough, malaise, diarrhea, dyspnea, and taste and smell impairment. The second hypothesis would be that this headache could be attributed to viral encephalitis or meningitis. However, it remains uncertain whether the headache from COVID-19 is related to direct viral damage to the peripheral or central nervous systems. Also, the number of patients with encephalitis, meningitis, or encephalopathy infected with SARS-CoV-2 kept increasing [13].

There is also a link between headache and rhinosinusitis in the case of COVID-19. In approximately half of the patients infected with SARS-CoV-2, rhinosinusitis appears as a symptom. Thus, in most cases, the headaches in these patients are closely related to acute rhinosinusitis (ARS). In some situations, the headache occurs without acute ARS; thus, rhinosinusitis can be encountered in the case of SARS-CoV-2 infection, but it does not appear in a certain way [13].

Globally, as of 12:15 pm CEST, 21 June 2023, there have been 768,187,096 confirmed cases of COVID-19 (211,331 new cases in the last 7 days) and 6,945,714 cumulative deaths reported to WHO. Until 21 June 2023, reported to the WHO were 276,545,765 confirmed cases in Europe, 204,478,043 confirmed cases in the Western Pacific, 193,056,651 confirmed cases in the Americas, 61,185,070 confirmed cases in South East Asia, 23,382,124 confirmed cases in the eastern Mediterranean, and 9,538,679 confirmed cases in Africa. As of 19 June 2023, a total of 13,461,344,203 vaccine doses had been administered. With the appearance of variants such as Omicron, the cases of COVID-19 increased exponentially, resulting in thousands of deaths [14].

## 3. Treatments for COVID-19

Treatments for COVID-19 include antivirals (e.g., molnupiravir, remdesivir, Paxlovid), anti-inflammatory drugs (e.g., dexamethasone), immune modulators (e.g., tocilizumab, baricitinib), and anti-SARS-CoV-2 monoclonal antibodies (e.g., casirivimab/imdevimab, bamlanivimab/etesevimab) [15,16].

Regarding antiviral treatments, Remdesivir is the one recognized by the FDA for the treatment of patients infected with SARS-CoV-2. Other antivirals, such as lopinavir and ritonavir (used to treat HIV infection), have been used as therapeutic agents against COVID-19. However, according to some reports, no significant improvements have been observed in patients infected with this virus. Chloroquine is another drug whose sulfate and phosphate salts (used in malaria) have been touted against SARS-CoV-2, but eventually not found to be effective. Molnupiravir is another antiviral drug that was recently approved by the FDA against infection with the SARS-CoV-2 virus [17].

Two monoclonal antibodies, Tocilizumab and Sarilumab (used in rheumatoid arthritis), were recommended for patients in the ICU by the NHS [18].

Casirivimab, together with Imdevimab, form another monoclonal antibody complex called REGEN-COVT, which has been reported to reduce hospitalization and death in COVID-19. This treatment is approved by the FDA for patients with mild and moderate symptoms, as well as for patients (both children and adults) with severe symptoms [19].

The most successful approach against COVID-19 has been vaccination, which has been available to the public since late 2020 with BNT162, developed by Pfizer and BioNTech and first rolled out in the UK. Various other variants are approved, as illustrated below [20].

The mRNA (messenger RNA)-based vaccines are composed of chains of messenger ribonucleic acid (mRNA) encapsulated in lipid nanoparticles. The first mRNA vaccines to receive emergency use authorization were BioNTech/Pfizer and Moderna. The effectiveness of those developed by Moderna and BioNTech/Pfizer is almost 95%. They encode the viral glycoprotein S (spike) of the SARS-CoV-2 virus [20].

Viral vector-based vaccines use viruses to carry genes that encode vaccine antigens into host cells. The vector is a virus other than the one targeted by the vaccine (e.g., an adenovirus). The genes of a pathogen are first put into the genome of a viral vector [21].

Vaccines with viral vectors can be classified into two types: replicable and non-replicable. Vaccines based on viral vectors that can replicate infected cells not only lead to the production of antigens but also to the reproduction of the viral vector (and hence, subsequently, the amplification of the immune response). Several COVID-19 vaccines are based on this technology: Oxford-AstraZeneca, Johnson & Johnson, and Sputnik [21]. The Johnson & Johnson vaccine (JNJ-78436725) uses a human adenovirus, Ad26, as a viral vector encoding a variant of the SARS-CoV-2 protein S [22].

Sinovac or CoronoVac (produced in China and Brazil) is a classical vaccine in the sense that it uses the inactivated virus as an antigen. Its efficacy against mild or moderate forms of COVID-19 has been reported to be distinctly lower than that of the mRNA or of the viral vector vaccines [22].

A fourth approach has been taken by the NVX-CoV2373 vaccine (made in the USA), using only protein subunits of the virus—but still with lower efficacy than the mRNA and viral vector versions [23].

According to the COVID-19 Treatment Guide Panel, regarding the treatment of COVID-19 until June 2023, here are the following conclusions:
Remdesivir is the only antiviral drug approved by the FDA for patients infected with SARS-CoV-2 who are older than 28 days and weighing more than 3 kg. No clinical drug–drug interaction studies of RDV have been performed. In the case of patients infected with SARS-CoV-2 with an eGFR < 30 mL/min, the FDA does not recommend the use of RDV.Ritonavir-Boosted Nirmatrelvir (Paxlovid) was authorized under an FDA EUA in COVID-19 for mild-to-moderate symptoms in high-risk patients over 12 years of age who weighed over 40 kg. Regarding drug–drug interactions in the case of nirmatrelvir stimulated with RTV, they are very important. It is recommended to carefully check the drugs that are still being administered to the patient, including herbal supplements, OTC drugs, or even recreational drugs.Molnupiravir has been authorized under the FDA EUA for COVID-19 in patients with mild to moderate symptoms in high-risk patients over 18 years. In the case of MOV, no clinical studies have been conducted for the drug–drug interaction.High-Titer COVID-19 Convalescent Plasma was authorized under the FDA EUA in COVID-19 in the case of immunocompromised patients or for patients under immunosuppressive treatment. It is not recommended to add it to the intravenous infusion. It is also recommended to decrease the CCP volume or the transfusion rate in patients with heart failure or cardiac dysfunction.IFN Beta was not approved by the FDA as it is still in clinical trials.PEG-IFN Lambda Beta was not approved by the FDA as it is still in clinical trials [24].

The COVID-19 vaccines have been approved by the Food and Drug Administration (FDA), meeting the standards required by them regarding the effectiveness and safety of the patients to whom they are administered [25].

According to the indications given by the CDC, for children between the ages of 6 months and 5 years, it is recommended to receive several doses of the vaccine, according to certain criteria: the number of doses previously administered, age, etc. For those aged between 5 months and 4 years, the administration of three doses of the Pfizer-BioNTech COVID-19 vaccine with at least one updated dose is recommended, and for those aged over 5 years, at least one updated dose of the vaccine is recommended. In the case of the Moderna vaccine, it is recommended to take the two doses, with at least one updated dose. For children over 6 years of age, the updated Pfizer-BioNTech or Moderna vaccine is recommended. The elderly, over 65, are recommended to take the second dose of Pfizer-BioNTech or Moderna. Those who are immunocompromised are advised to take additional doses of the updated Pfizer-BioNTech or Moderna vaccine. For patients who choose not to receive an mRNA-based vaccine, depending on age and dose recommendations, Novavax or Johnson & Johnson/Janssen vaccines with an updated dose are available [25].

In the case of COVID-19, as far as allergies are concerned, there are few situations, many of which are only immunogenic reactions not mediated by IgE. However, there are situations in which it is recommended to avoid them in order to avoid side effects that could sometimes be dangerous, or even fatal. Two important potential allergens in the case of COVID-19 vaccines are PS80 and PEG. PS80 (polysorbate 80) is a polymer derived from polyethoxylated sorbitan and oleic acid and is a potential allergen in Janssen/Johnson & Johnson and Astra Zeneca vaccines. It is used in many other vaccines and medicines as an excipient and is also a food additive. PEG is a polymer of ethylene oxide. It is found as an excipient in the Pfizer/BioNTech and Moderna vaccines and is not indicated for patients with PEG allergies [26,27].

The two excipients have a similar chemical structure, both being polymers derived from ethylene oxide, hence the possibility of cross-reactivity between PS80 and PEG. To date, there are no COVID-19 vaccines that do not contain PS80 or PEG. To avoid anaphylaxis in the event of the administration of one or more doses of the COVID-19 vaccine, an algorithm has been created. Thus, the first dose can be administered in the case of patients who have no history of anaphylaxis to PEG or other drugs. In the situation where the patient, due to anaphylaxis, did not tolerate PEG, it is recommended to avoid the administration of vaccines containing PEG, the most suitable option being those containing polysorbate. In the case of the second dose, if after the administration of the first dose containing PEG no side effects appeared, the second dose can be administered. If after the first dose there was a slight urticaria or a delayed systemic reaction located at the injection site and the reactions could be easily resolved with antihistamines, the second dose can be administered after the overdose with cetirizine 20 mg. It is recommended to monitor the post-vaccination symptoms for a longer period; in case they become more severe, further investigations are necessary. If the reactions in the first 2 h after the administration of the first dose containing PEG were severe (severe systemic allergies, anaphylaxis), the COVID-19 vaccine is not administered. Skin tests will be performed for the available vaccines for PEG and other allergens, and then, under longer and more careful supervision, the administration of a COVID-19 vaccine that matches according to the skin tests can be taken into account [27].

## 4. Long-Term Effects of SARS-CoV-2

Over 100 million people around the world have recovered from the infection with COVID-19. However, in some of these cases, severe symptoms and organ dysfunctions were found to persist. These long-term symptoms have been proposed to be linked to the cytokine storm that occurs in the acute phase of the infection—especially as a similar correlation has also been reported in two other coronavirus infections—MERS (Middle East respiratory syndrome) and SARS (severe acute respiratory syndrome). However, while in those two cases the severe effects were reported to be concentrated around lung infections, COVID-19 does not appear to be just a respiratory condition, and SARS-CoV-2’s long-term effects appear to affect multiple systems/organs [28].

According to one study, over 50 long-term effects/symptoms have been reported for SARS-CoV-2. Most of these are clinical symptoms also observed during the acute phase, e.g., headache, fatigue, ageusia, anosmia, and joint pain. In addition, illnesses such as diabetes and stroke were also present. Among these, the five most frequent manifestations were headache (44%), fatigue (58%), attention disorder (27%), dyspnea (24%), and hair loss (25%). Other symptoms have been correlated with pulmonary (chest discomfort, cough, sleep apnea, reduced pulmonary diffusing capacity, and pulmonary fibrosis), neurological (depression, dementia, anxiety, obsessive-compulsive disorder, attention disorders), and cardiovascular (arrhythmias, myocarditis) systems, while others were less specific, e.g., tinnitus, hair loss, and night sweats [29].

More neuropsychiatric symptoms were reported: attention deficit disorder (27%), headache (44%), and anosmia (21%). These have been proposed to either be due directly to the infection, to side effects of the treatment, or to hypoxia or cerebrovascular disease (including hypercoagulation). Adults have a bigger risk of being diagnosed with a psychiatric illness after COVID-19 than children [29].

Cough and dyspnea were encountered in 19% and 24% of patients, respectively. Abnormalities in lung CT scans were noted in 35% of patients even after 2–3 months from initial presentation. According to a study on non-critical patients, radiographic changes persisted in nearly 2/3 of patients at 3 months after discharge [30].

An association between diabetes and COVID-19 has been observed, which could be attributed to the impact of the SARS-CoV-2 infection on the pancreas [29]. Following SARS-CoV-2 infections, a reduced number of insulin-secreting granules in beta cells and thus a reduction in insulin secretion were observed. SARS-CoV-2 has been reported to destroy beta cells by triggering pro-inflammatory cytokines. In adipose tissues, proinflammatory pathways leading to chronic low-grade inflammation are very important for type 2 diabetes and the pathogenesis of insulin resistance. Cases of hyperglycemia and insulin resistance have been reported in COVID-19 patients without a history of diabetes [31].

As a result of the SARS-CoV-2 infection, the body releases a wave of cytokines [32]. In some cases (possibly due to genetic differences), this common defensive event is extreme, creating vulnerability and weakening of the immune system and causing inflammation that can devastate the body, severely affecting organs such as the liver, kidneys, and heart. Although COVID-19 is primarily a respiratory disease, the heart can also be affected. Permanent or temporary injury of the heart tissue can be due to some factors, such as lack of oxygen, myocarditis (a coronavirus can directly infect and damage heart muscle tissue), or stress cardiomyopathy (cardiomyopathy can occur as a consequence of viral infections and is a disorder of the heart muscle that changes the heart’s capacity to pump blood effectively) [9,33].

The thyroid cells contain larger amounts of the ACE-2 receptor and are hence fragile against SARS-CoV-2 infection [34]. Two hypotheses were proposed by which the thyroid function of patients with COVID-19 is affected: a direct one and an indirect one. The direct one considers the effect of SARS-CoV-2 on target cells, and the indirect one is related to the inflammatory immune response. A third proposed mechanism is the fact that HPT axis (pituitary-thyroid axis) dysfunction causes a decrease in the serum level of TSH in infected patients in the acute phase of the infection. In any event, apoptosis in SARS-CoV-2-induced thyroid lesions has been observed [35].

In the case of patients with COVID-19, a decrease in the plasma level of antioxidant enzymes such as glutathione peroxidase (GPx), glutathione (GSH), superoxide dismutase (SOD), and catalase was observed, as well as increases in oxidative stress parameters, thus increasing the risk of severe disease and mortality. Oxidative stress in this case is influenced by the production of inflammatory cytokines, the activation of the innate immune response, or the death of infected cells. In the case of patients with cataracts, a decrease in the serum concentration of antioxidant total sulfhydryl (SH) groups and other enzymatic antioxidants and, at the same time, an increase in the level of malondialdehyde were observed. A connection between total proteins and the level of markers of oxidative stress and the worsening of cataracts was also noticed in the aqueous humor. According to some research, in COVID-19, the lenticular proteins, nucleic acids, and lipids are more easily oxidized due to the degradation of the antioxidant response and also the decrease in the efficiency of the repair mechanisms in the cataract lenses. Another important factor in the progression of cataracts in COVID-19 is the administration of corticosteroids in some patients, probably due to oxidative damage, osmotic imbalance, inhibition of lens growth factors, and disruption of connective tissue metabolism. Dehydration, malnutrition, increased urea levels, and diarrhea are also factors that can lead to a reduction in GSH levels, with increased cyanate levels in COVID-19 also being a cataractogenic factor [36].

## 5. Comorbidities

Numerous risk factors have been reported for COVID-19; the most commonly noted are gender (for males), older age, and pre-existing comorbidities [37].

As pointed out above, SARS-CoV-2 affects not only the lungs, but also other targets—especially the heart, the brain, and the gastrointestinal system. According to some statistics, it was observed that 75% of hospitalized patients with COVID-19 have at least one comorbidity. The most common among these are hypertension, diabetes, cancer, neurodegenerative diseases, cardiovascular diseases, obesity, and kidney diseases. A study among 99 patients reported that 50 of these suffered from chronic diseases: diabetes (12%), cardiovascular diseases (40.4%), diseases of the digestive system (11%), and malignant tumors (0.01%). Other studies have also reported comorbidities that frequently increase the risk of mortality in patients, such as lung cancer, chronic myeloid leukemia, and infections with the HIV virus [38,39,40].

Comorbidities have also been observed to have similar effects in other respiratory diseases, such as MERS [38].

As pointed out above, SARS-CoV-2 relies on a high affinity for the receptor ACE-2, which is present in large amounts not only in the epithelial cells of the lungs, but also in the intestine, kidneys, and blood vessels. Notably, certain treatments used for various other diseases (in other words, comorbidities) can increase the expression of ACE-2 and hence increase the likelihood of infection with SARS-CoV-2 and of more severe symptoms therefrom [37].

A large number of articles indicate that infectious diseases may affect men and women differently. Estrogen can potentiate the immune activities of vitamin D, hence having the potential to improve the clinical outcomes of COVID-19 infections. By contrast, it has been proposed that male sex hormones increase the activity of the ACE-2 receptor and hence increase the susceptibility to SARS-CoV-2. On a different note, testosterone has been proposed to exert immunosuppressive effects, hampering the antibody response [41].

Another risk factor correlated with the severity and poor clinical results of infection with COVID-19 is age. This may be linked simply to the general state of the immune system in older people. Not only have children been reported to be more often asymptomatic than adults (as pointed out above), but they are also less likely to develop severe forms of COVID-19. In contrast, elderly patients are more likely to develop severe symptoms and have higher death rates. The comorbidities present in the elderly are thought to be the main reason for this situation. However, the age and the real state of health of a patient are independent variables in the correlation with serious symptomatology in the case of SARS-CoV-2 infection [41].

### 5.1. Meta-Analyses on Diabetes

As described above, diabetes is among the most common comorbidities in COVID-19 patients [42]. Based on clinical studies, it appears to be correlated with severe complications that include acute respiratory failure syndrome, pneumonia, multiple organ failure, or death [43]. A study carried out on 5700 COVID-19 patients from 12 hospitals in the USA found diabetes to be the third most common comorbidity (~34% of patients), compared to hypertension (56%) and obesity (42%) [44]. Therapeutically, intensive insulin therapy for normalization of blood glucose levels and suppression of ketosis was found to significantly reduce morbidity and mortality in COVID-19 patients with diabetes [43].

Diabetes was found to be associated with a more than two-fold higher risk of severe COVID-19 disease and a three-fold higher risk of mortality compared to COVID-19 patients without diabetes. Diabetic ketoacidosis has been cited as a key aggravating factor in these instances [45].

Diabetic ketoacidosis is four times more common in type 2 diabetes than in type 1. Further stress on the organisms (e.g., surgery, trauma, infections—including COVID-19) increases the risks of ketoacidosis. The majority of diabetes cases are type 2 (~90%) accounting for ~9% of the adult population [46].

In a study involving 639 cases, the majority of patients with diabetic ketoacidosis diagnosed with COVID-19 were adults (52%), men (58%), and had pre-existing type 2 diabetes (48%). A decrease in the odds of survival was correlated with type 2 diabetes but not with type 1 diabetes. Severe disease was noted to almost always be the direct cause of ketoacidosis and subsequent death in type 2, but not in type 1 diabetes [47]. Another determining factor of mortality in COVID-19 patients with diabetic ketoacidosis identified in this study could be the low pH (pH <  7) compared to those who survived. A high level of uncontrolled blood glucose (>1000 mg/dL) was also found to be correlated with a decreased chance of survival [47]. It was also discovered that of the 11 cases of diabetic ketoacidosis (1.7%) who were taking sodium-glucose cotransporter-2 (SGLT-2) inhibitors, 18.2% (two patients) were diagnosed with euglycemic ketoacidosis associated with SGLT2 (blood sugar < 250 mg/dL) [47]. Because of their renal and cardiovascular benefits, SGLT-2 inhibitor drugs are often used in the treatment of patients with type 2 diabetes. There are now suggestions that the use of these drugs should be discontinued in COVID-19 patients if they develop ketosis [48].

Insulin resistance is accompanied by excess protein glycation and increased oxidative stress, followed by a general inflammatory response (e.g., with cytokines and adhesion molecules), which increases vulnerability to infection. In turn, the infection can affect the pancreatic beta cells (also a site of ACE-2 receptors), further limiting the ability to produce insulin and amplifying the patient’s vulnerability. SARS-CoV-2 has been reported to destroy the islet cells to the extent of favoring the onset of diabetes [49].

A recent meta-analysis of 2108 COVID-19 patients found the frequency of diabetes to be 10% with a mean age of 49 [50].

### 5.2. Meta-Analyses on Cardiovascular Diseases and Hypertension

#### 5.2.1. COVID-19 and High Blood Pressure (HT)

High blood pressure (HT) affects approximately 30% of adults worldwide and represents a risk factor for a number of conditions that include stroke, cardiovascular disease, chronic kidney disease, and premature death [51]. Arterial hypertension is one of the most common comorbidities associated with COVID-19; however, it is unclear to what extent hypertension in itself is an aggravating factor or if it is predominantly an inherent co-variable associated with advanced age and the weakened immune system therefrom [52].

The renin-angiotensin-aldosterone system (RAAS) has an important role in blood pressure (BP) regulation via two axes, one of which involves the enzyme ACE-2. The interaction of SARS-CoV-2 with this receptor is then reasonably expected to destabilize RAAS [51].

A meta-analysis showed an association between HT and the severity and mortality of COVID-19. The likelihood of death was increased by ~3.5 times by HT, and the likelihood of severe symptoms by ~2 times—with slightly stronger effects in patients older than 50 [53].

In one more study, hypertension in COVID-19 patients correlated with an increased mortality rate (~25% vs. ~15%), the occurrence of severe symptoms (~634% vs. 42%), the requirement for mechanical ventilation (~17% vs. ~8%), and an increased likelihood of transfer to the ICU (~23.4% vs. ~12%) [52].

Another study involving 6560 cases showed that in COVID-19 patients, hypertension is an important factor in increasing the severity of infection-induced pneumonia and mortality [54].

#### 5.2.2. COVID-19 and Cardiovascular Diseases

Cardiovascular diseases (CVD) are, as discussed above, generally associated with the severity of COVID-19. Documented cases in this class of co-morbidities include myocardial infarction, coronary heart disease, and heart failure [52].

In one study, CVD was correlated with increased mortality in COVID-19 patients: 13 of 68 in the CVD group vs. 0 of 82 in the control group [55]. Another study showed that CVD also increases the likelihood of severe symptoms—from ~9% to ~28% [56]. A meta-analysis concluded that CVD brings about an increase in mortality risk by ~5 times in COVID-19 patients [57].

Cardiac injury is also a relatively common symptom (not only/simply a comorbidity) in people infected with SARS-CoV-2. Some studies have reported increased mortality and morbidity in patients with pre-existing myocarditis. Patients with heart damage were more likely to require mechanical ventilation than those without heart damage [52].

Mechanisms by which SARS-CoV-2 can cause cardiovascular disease have been suggested. Direct infection of myocytes via ACE-2 has been proposed as a major factor—although a role for the cytokine storm has also been suggested—via the ensuing inflammation, arrhythmia, and erosion of coronary plaques [58].

### 5.3. Meta-Analyses on Obesity

Obesity is well known to be a risk factor for cardiovascular disease and diabetes. In addition, many respiratory complications have been associated with obesity, including increased ventilatory demand, respiratory muscle inefficiency, and decreased respiratory compliance. Compared to China, obesity is more prevalent in Italy, which may contribute to the numerous deaths. Another point to note is that the United States, which has the highest death rates from COVID-19, has a high frequency of obesity compared to China. All of these observations have raised concerns about the impact obesity may have on COVID-19. It has been proposed that obesity may inherently entail upregulation of cell membrane receptors that allow SARS-CoV-2 to attach and enter the cell [59]. Indeed, obesity increases the expression of ACE-2 in adipose tissue. Secondly, obesity is also known to decrease the efficiency of the immune system and increase the general likelihood of inflammation, which can have effects on the lung parenchyma and bronchi. Adipose tissue secretes adipokines (cytokines with roles in inflammation and metabolism) and components of the SRA (renin-angiotensin system), all of which act on subsequent targets such as the brain and the metabolic and immune systems. Last but not least, obesity can also reduce the capacity of the lungs, and pose more difficulties for mechanical ventilation [60,61].

According to a study carried out on 150,000 US adults infected with SARS-CoV-2, it was observed that half of these patients were obese and approximately 28% were overweight. Among these, the risk of hospitalization, ICU admission, or death showed correlations with body-mass index values (BMI) [62].

Out of eleven studies, ten showed an increase in mortality in COVID-19 patients who were overweight, obese, or had severe obesity [29,34,44,45,46,47]. The eleventh study did not observe any difference between overweight patients (BMI > 28) and those of normal weight. However, an increase in the severity of symptoms was observed in overweight patients. Since BMI poorly reflects the distribution of adipose tissue throughout the body, four of the studies used VAT (visceral adipose tissue) obtained by high-resolution computed tomography as a measurement. In COVID-19 presented at emergency departments, subsequent ICU admission was correlated with a 30% higher VAT but also with a 30% lower subcutaneous adipose tissue (SAT). Another study also observed no BMI effect but did find correlations between increased VAT, a visceral fat test/total adipose tissue (TAT), and the need for hospitalization. In another study, VAT and TAT were again found to be associated with ICU admission [60].

Vitamin D plays an important role in the case of COVID-19 because the patient’s metabolic alteration is very likely, which leads to an increase in the risk of severe disease. Among the mechanisms from which the low level of vitamin D concentration in the form of 25(0H) D in obese patients can be deduced are vitamin D isolation in adipose tissue, volumetric dilution, or a negative 1,25-dihydroxy vitamin D response. Vitamin D has anti-inflammatory, immunomodulatory, and antiviral effects by regulating the level of inflammatory cytokines and decreasing the number of leukocytes in filtration. One way in which this is possible would be to increase the concentration of cathelicidins and defensins, which have an important role in decreasing the concentration of pro-inflammatory cytokines, a fact that leads to a decrease in the risk of a fatal cytokine storm in some patients [12].

In addition, another important factor in obese patients who develop the disease in the case of SARS-CoV-2 infection is intestinal microbiome dysbiosis and endotoxemia. Endotoxin, or LPS (lipopolysaccharide), is a glycolipid found on the outside of the membrane of Gram-negative bacteria. It is made up of three structural parts: lipid A, which is lipophilic; polysaccharides or oligosaccharides, which are hydrophilic; and antigen O. The structure of lipid A (the number of acyl groups) determines its role as an immunostimulator. The intestinal microbiome consists of all the microorganisms in the gastrointestinal tract. The human one contains two important components: Firmicutes (mainly Gram-positive bacteria) and bacterial phyla—Bacteroidetes (the Gram-negative bacteria). According to some studies, obese people have a higher level of Firmicutes compared to Bacteroidetes, which are lower. Thus, in the case of some people with obesity, due to the dysbiosis of the intestinal microbiome, the intestinal LPS composition could be modified in favor of some pro-inflammatory LPS subtypes because phyla—Bacteroidetes—will produce a greater amount of inflammatory LPS. Lipid A is the one that initiates a cascade resulting in the activation of some pro-inflammatory pathways, especially NF-κB and at the same time an increase in oxidative stress in the context of TLR4 binding. Also, a link between LPS and glycoprotein S on the surface of SARS-CoV-2 and the production of a cytokine response and NF-kB in the blood and in monocytes was noticed. In conclusion, by limiting the caloric intake of saturated fats and trans-fats and also by regulating the intestinal microbiota (with synbiotics, probiotics, prebiotics, etc.), endotoxins could be reduced with a decrease in the risk of severe forms in the case of obese patients infected with SARS-CoV-2 [63].

### 5.4. Meta-Analysis on Cancer

In cancer, the immune system is weakened by the tumor itself as well as by many of the anticancer treatments, which leads to an increased risk of infection and worse symptoms therefrom in general and with SARS-CoV-2 in particular. Additionally, cancer patients also tend to be older than the general population, which, as discussed before, is also a risk. A two-fold increase in the risk of SARS-CoV-2 infection was thus reported for cancer patients over the general population. The frequency of COVID-19 is 2–3% in cancer patients, with more severe symptoms in these patients that can lead to hospitalization in the ICU, invasive ventilation, and even death compared to normal people. According to some meta-analyses, malignant tumors were correlated with worse clinical outcomes [64].

In a meta-analysis of 15 studies on patient groups that were very diverse geographically and ethnically, 14 showed a mortality rate of ~22% in the case of patients diagnosed with COVID-19—approximately 4 times higher than the rate in patients without cancer within the respective groups [64]. In the same study, a series of subgroup analyses were performed regarding the rate of mortality or severe symptoms according to the type of cancer and the type of treatment. The mortality rates in COVID-19 patients with lung cancer were very similar to those in patients with hematological cancer (33–34%), which was approximately double compared to the rates seen in patients with other types of cancer. The incidence of severe events was, however, 55–60% regardless of the type of cancer [64]. The type of cancer treatment received by these patients did not correlate clearly with death rates or adverse effects [64].

### 5.5. Meta-Analyses on Renal Disease

A meta-analysis of 76,993 patients noted chronic kidney failure (CKD) among the seven most common comorbidities in hospitalized COVID-19 patients [65].

A study conducted on 701 patients with COVID-19 found increased frequencies of elevated kidney-related markers (elevated urea nitrogen 13%, elevated serum creatinine 14%, estimated glomerular filtration 13%), alongside acute kidney injury in ~5% of the patients. A meta-analysis indicated CKD to be correlated with severe symptoms in COVID-19 [66].

There are many reasons why the disease COVID-19 could give rise to kidney failure: the nephrotoxicity of certain treatments, the symptoms of COVID-19 themselves, hypoxia (insufficient supply of oxygen to the tissues), the virus-induced cytokine storm, and mechanical ventilation. Importantly, older age and other comorbidities (e.g., hypertension, diabetes, chronic cardiovascular disease, or chronic liver disease) further increase the likelihood of secondary kidney diseases [66].

Other studies have also reported renal disfunction in COVID-19: 63% of patients developed proteinuria, 34% had massive albuminuria, and 27% had the nitrogenous portion of urea (BUN) increased [66]_._ Another study, performed on 710 hospitalized patients infected with COVID-19, reported 27% hematuria, 44% proteinuria, and 14% BUN. Renal inflammation, edema, and reduced density were confirmed in these cases by CT imaging [67]. Regarding patient prognosis, of 443 COVID-19 patients, ~19% had CKD—distinctly larger than the general population [66].

Another study reported that of 4264 patients with CKD, mortality rates were up to 50%, and this was not affected by whether the patients were on dialysis or not—though dialysis was found to be correlated with a higher risk of contracting COVID-19 [68].

### 5.6. Meta-Analyses on Liver Disease

MERS and SARS-CoV (which have a genome close to SARS-CoV-2) had an association with severe disease (with increased levels of transaminases in the blood) and liver damage, and a similar situation was thus to be expected in COVID-19 [69].

In COVID-19, liver damage could be observed in some cases. However, there is no clear proof regarding the clinical significance, and at the same time, it is still not clear if pre-existing chronic liver disease (CLD) is a factor in COVID-19 [70].

Among the main possible mechanisms of liver injury in COVID-19 are psychological stress, the lesions produced by the virus due to the ACE2 receptor, medication, pre-existing liver diseases, the systemic inflammatory response, and others [70].

According to a meta-analysis of patients with COVID-19, CLD did not increase the risk of infection. However, CLD was found to correlate with a higher risk of having more severe symptoms and a subsequent increase in mortality [71].

Preexisting infection with the hepatitis B virus was not found to correlate with severe or critical symptoms. In the case of these patients, a decrease in blood monocytes and a significant increase in CD8-T cells were observed, as well as an increased risk of thrombocytopenia, higher prothrombin, and hence a risk of abnormal coagulation [72]. By contrast, in the case of patients with active hepatitis C virus, more severe symptoms were observed following infection with SARS-CoV-2 [69].

Non-alcoholic fatty liver disease (NAFLD) was present in the majority of COVID-19 patients with CLD. Patients with NAFLD were found to have a longer viral shedding time and to be more prone to liver dysfunction and severe COVID-19 symptoms. In patients with NAFLD and older (>50 years), ALT elevation was observed in COVID-19. Likewise, NAFLD with increased values of FIB-4 or NFS was found to correlate with more severe COVID-19 symptoms [73].

Another important change consisted of increases in ALT, AST, and, in a few cases, jaundice. Other studies have indicated, following the autopsy of some patients infected with COVID-19, a dilation of the endoplasmic reticulum, apoptosis of hepatocytes, and a decrease in glycogen, among other necrosis. Data from other autopsies were interpreted as evidence that these changes were caused by the cytokine storm and/or by pre-existing liver diseases [74].

The strongest factor for developing severe disease in COVID-19 patients with liver disease was observed in those with cirrhosis and autoimmune hepatitis (AIH). An increase in mortality was observed in patients with COVID-19 who have cirrhosis as a comorbidity, compared to patients who only have COVID-19 (30% vs. 13%) [75].

### 5.7. Meta-Analyses on Gastrointestinal Disease

After infection with SARS-CoV-2, gastrointestinal dysfunctions were also very often encountered. Thus, inflammatory bowel disease (IBD) has become a worrying aspect in COVID-19 patients, in this case being a greater risk for severe disease [76]. In IBD, an important role is played by age, location of the disease, and inflammation. In the terminal ileum, a higher ACE-2 expression was observed than in the colon [77]. On the other hand, a meta-analysis found no increased risk of infection or severe disease in COVID-19 patients with IBD [78]. Patients who received treatments and therapies such as anti-TNF, JAK inhibitors, vedolizumab, and ustekinumab did not show an increased risk of infection with COVID-19 [78].

While according to some studies, patients with IBD who have contracted SARS-CoV-2 have an increased risk of developing severe disease and increased mortality, it has been argued that these findings are correlated with the older age of the respective patients, which, in turn, entails increased chances of comorbidities that may include active IBD [79].

Studies have been conducted to evaluate the risk factors in IBD patients who were infected with SARS-CoV-2, studies that focused on the regulation of ACE-2 and TMPRSS2 in the ileum and colon and on monitoring the effects of drugs used for IBD regarding the change in the expression of these two proteins and the increase in the vulnerability of the disease. It was observed that the location of the disease, the age, and the activity of the disease are decisive factors; an increase in ACE-2 in the disease located in the colon and an increase in TMPRSS2 in the disease located in the ileum were noted [77,79].

## 6. Conclusions

COVID-19 presents a wide spectrum of manifestations with poor clinical results or even mortality in vulnerable, elderly people and with pre-existing comorbidities, such as: hypertension, diabetes, obesity, cardiovascular disease, chronic kidney disease, cancer. The findings are summarized in Table 1:

Thus, it is important that patients with one or more of the comorbidities listed above take the necessary precautions to avoid, if possible (including, importantly, with vaccination), infection with SARS-CoV-2 in order not to reach more severe complications or even death. The summary offered here also provides a synthetic starting point for professionals in healthcare in terms of adjusting their approach towards special groups of patients. Such an overview is expected to also be of interest for understanding the variability of symptoms and hence the mechanisms whereby special groups may exhibit peculiar responses to the disease—all of which may be important in developing better treatments not only for COVID-19 but also for related viral diseases, especially those that may arise in the future.

## Figures and Tables

**Table 1 medicina-59-01393-t001:** The effects of SARS-CoV-2 infection and the mechanisms proposed in different comorbidities.

Comorbidity	Effects	Proposed Explanation/Mechanism
Diabetes	- Respiratory failure syndrome - Pneumonia - Multiple organ failure - Death	- Ketoacidosis in type 2 diabetes - High level of uncontrolled blood glucose -> decreased chance of survival - Side-effects of sodium-glucose cotransporter-2 (SGLT-2) inhibitors - Infection affects/destroys pancreatic beta cells -> impaired insulin production
Cardiovascular diseases	- Myocardial - Infarction - Coronary heart disease - Heart failure - Increased chance of mechanical ventilation in patients with heart damage	- Direct infection of myocytes via ACE-2 - Cytokine storm leads to inflammation, arrhythmia, and the erosion of coronary plaques
Hypertension	- Stroke - Cardiovascular disease - Chronic kidney disease - Premature death - Requirement for mechanical ventilation - Increased likelihood of transfer to ICU	- Inherent co-variable with advanced age and weakened immune system - virus interaction with ACE-2 destabilizes the RAAS (renin-angiotensin-aldosterone system)
Obesity	- Increased ventilatory demand - Respiratory muscle inefficiency - Decreased respiratory compliance - VAT and TAT—associated with ICU admission	- Increased adipocytes ACE-2 expression - Reduced efficiency of the immune system, increased chance of inflammation - Adipokines and components of the SRA (renin-angiotensin system) produced by adipose tissue are affected by the virus - Reduced lung capacity of the lungs, difficulties for mechanical ventilation - Decreased vitamin D bioavailability - Gut microbiome dysbiosis
Cancer	- More severe symptoms, increased likelihood of ICU - Invasive ventilation - Death - Malignant tumors -> worse clinical outcomes - Doubled mortality rates in COVID-19 patients with lung or hematological cancer vs. other types	- Weakened immune system, caused by the tumor as well as by many anticancer treatments - Cancer patients tend to have an older age than the general population
Renal disease	- Elevated urea nitrogen, serum creatinine, total protein, albumin, BUN, glomerular filtration - Acute kidney injury - Renal inflammation, edema, reduced density - Dialysis correlates with higher risk of infection	- Nephrotoxicity of certain treatments - Hypoxia - Virus-induced cytokine storm - Mechanical ventilation - Older age and other comorbidities (e.g., hypertension, diabetes, chronic cardiovascular, or chronic liver disease) increase the likelihood of secondary kidney diseases
Liver disease	- Increased transaminases, prothrombin; CD8-T; decreased blood monocytes - Liver damage - Risk of thrombocytopenia - Abnormal coagulation - Patients with NAFLD—longer viral shedding time, liver dysfunction and severe symptoms	- Psychological stress - Virus binding via ACE-2 receptor - Medication - Systemic inflammatory response - Dilation of the endoplasmic reticulum, apoptosis of hepatocytes, decrease in glycogen, possibly due to cytokine storm and/or by pre-existing liver diseases - Strongest effects: in cirrhosis, autoimmune hepatitis
Gastrointestinal disease	- Increased risk of severe disease - Increased mortality	- ACE-2 at the colon and TMPRSS2 at the ileum - Correlation with older age and inflammation

## Data Availability

Not applicable.

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
