# Peer review of "Comorbidities of COVID-19 Patients"

_medicina, 2023, doi:10.3390/medicina59081393_

Round 1

Reviewer 1 Report (Previous Reviewer 2)

The authors have revised their manuscript entitled “Comorbidities in COVID-19”.  

The parts of the manuscript which are improved after peer-review are not mentioned in the manuscript. They should be mentioned either in different text-colour or in different manner.

Thus, further review of the revised manuscript is not possible in this case.

I am strongly recommending that the manuscript has to be revised based on the previous review.

Reviewer’s Comment:

1.     The authors should mention which portion they have modified based on review.

2.     There are still some typos and grammatical error which should be removed.

3.     The writing of the manuscript is not scientifically designed. The authors should follow sample manuscripts in this regard.

Author Response

The previous revision did include as attached file a Track-changes version of the manuscript showing the extensive changes, and a detailed account of those changes was given in our replies and Letter at that time; this also included the statement regarding search strategies. We are now submitting this revision showing the previously modified text in yellow. Grammar corrections were also performed as requested (not highlighted). New/modified references are shown in green.

Reviewer 2 Report (Previous Reviewer 3)

Dear Author(s),

Thank you for all the modifications made. I have only few minor modifications needed before publication.

After modifications that you made according to my previous extensive comments the quality of the manuscript improved at to the best of my knowledge will be ready for publication after this minor corrections stated below.

Reference 61 should be cited in full (data is missing on your list): Belančić A, Kresović A, Rački V. Potential pathophysiological mechanisms leading to increased COVID-19 susceptibility and severity in obesity. Obes Med. 2020;19:100259. doi: 10.1016/j.obmed.2020.100259. 

Include a refrence for the statement you used. The statement "Acute hyperglycaemia, hypoglycaemia, as well as hyperglycaemia post‐hypoglycaemia all produce oxidative stress and are followed by enormous production of inflammatory cytokines and by enhancement of the inflammatory/infectious process." is from: Belančić A, Kresović A, Troskot Dijan M. Glucagon-like peptide-1 receptor agonists in the era of COVID-19: Friend or foe? Clin Obes. 2021;11(2):e12439. doi: 10.1111/cob.12439.

Author Response

References added/modified as suggested.

Reviewer 3 Report (Previous Reviewer 1)

The authors have fully addressed my previous comments and have included everything that was suggested in the manuscript.

Author Response

The present revision includes two new references requested by Reviewer 2 (shown in green) and some yellow highlighting to show what text was added in the previous revision (at the express request of Reviewer 1).

Round 2

Reviewer 1 Report (Previous Reviewer 2)

I am recommending the acceptance of the manuscript.

This manuscript is a resubmission of an earlier submission. The following is a list of the peer review reports and author responses from that submission.

Round 1

Reviewer 1 Report

The review paper presented to me for review deals with an interesting and clinically important issue - comorbidities in COVID-19. The paper is written very honestly and in good, understandable language. The manuscript is divided into typical sections and the results of the analysis are described logically and clearly. 

However, several important points need to be completed and clarified:

1. it is a review paper and so the authors should present the methodology and scheme of searching for papers, the so-called search strategy - what keywords they used and how many articles they qualified for the final analysis

2. the discussion needs to be supplemented with headaches. In the influence of COVID-19 on the most common diseases (including comorbidities), a few sentences should be completed about the most common neurological disease in the world - migraine and headaches in general. The influence of SARS-CoV2 virus on the potential pathogenesis of headaches (https://pubmed.ncbi.nlm.nih.gov/35758225/), the atypical course (https://pubmed.ncbi.nlm.nih.gov/34119843/ and https://pubmed.ncbi.nlm.nih.gov/34541916/ and ) should be mentioned. 

Author Response

  1. it is a review paper and so the authors should present the methodology and scheme of searching for papers, the so-called search strategy - what keywords they used and how many articles they qualified for the final analysis

Reply: paragraph added as suggested.

  1. the discussion needs to be supplemented with headaches. In the influence of COVID-19 on the most common diseases (including comorbidities), a few sentences should be completed about the most common neurological disease in the world - migraine and headaches in general. The influence of SARS-CoV2 virus on the potential pathogenesis of headaches (https://pubmed.ncbi.nlm.nih.gov/35758225/), the atypical course (https://pubmed.ncbi.nlm.nih.gov/34119843/ and https://pubmed.ncbi.nlm.nih.gov/34541916/ and ) should be mentioned.

Reply: done as suggested.

Reviewer 2 Report

Authors have studied the effects of COVID-19 infection in patients with different comorbidities. They reviewed the common symptoms, available treatments and long term effects of COVID-19 infection. Though there’s no novelty up to this review.

Thereafter, the authors performed Meta-analyses on several comorbidities like diabetes, cardiovascular diseases, hypertension, obesity, cancer, renal diseases, liver disease, and gastrointestinal diseases. They enlisted their gained results in the Conclusion segment.

The manuscript doesn’t provide significant contribution.  

I am strongly recommending that the manuscript needs major revision.

I attach my comments regarding the review of the manuscript:

Reviewer’s Comment:

1. The authors should attach the novelty of their study. Also, the scientific significances behind their research should be mentioned.

2. The authors should vividly provide a conclusion segment about their overall study.

3. The authors should mention that how would their study help the future researchers.

4. There are a few typos and grammatical errors.

Author Response

Reviewer 2:

  1. The authors should attach the novelty of their study. Also, the scientific significances behind their research should be mentioned. 2. The authors should vividly provide a conclusion segment about their overall study. 3. The authors should mention that how would their study help the future researchers.

Reply: some editing of the Conclusions to this end has been performed.

  1. There are a few typos and grammatical errors.

Reply: a number of corrections have been made.

Reviewer 3 Report

Dear Author(s),

Thank you for your interesting manuscript regarding COVID-9. I have several minor suggestions to highlight, please correct/modify your manuscript according to further suggestions:

  1. Update epidemiological numbers regarding global COVID-19 cases and deaths
  2. Add in the text and table 1 two more pathophysiological mechanisms regarding the association between obesity and COVID-19 - decreased vitamin D bioavailability ( 10.1016/j.obmed.2020.100259 ) and gut microbiome dysbiosis ( 10.1016/j.obmed.2020.100302 )
  3. Please also add a short comment on potential ocular complications of SARS CoV-2, such as cataract progression ( https://www.psychiatria-danubina.com/UserDocsImages/pdf/dnb_vol33_noSuppl%2013/dnb_vol33_noSuppl%2013_424.pdf)
  4. Highlight that acute hyperglycaemia, hypoglycaemia, as well as hyperglycaemia post‐hypoglycaemia all produce oxidative stress and are followed by enormous production of inflammatory cytokines and by enhancement of the inflammatory/infectious process
  5. Please include primary endpoints from the most important and registration clinical trials for the most important antiviral medical agents. Consider to mention potential drug-drug interactions that prescribers need to be aware of.
  6. Provide safety data for the vaccines (overview of most important and rare specific AE). Also mention which vaccines have polysorbate 80 and which PEG, which is important from the alergic point of view. Comment on booster doses of individual vaccines and associated efficacy.
  7. Conclusion need to be more specific and please extend it.

Not applicable. English is fine, only minor corrections are needed.

Author Response

Reviewer 3

  1. Update epidemiological numbers regarding global COVID-19 cases and deaths

Reply: done as suggested.

  1. Add in the text and table 1 two more pathophysiological mechanisms regarding the association between obesity and COVID-19 - decreased vitamin D bioavailability ( 10.1016/j.obmed.2020.100259 ) and gut microbiome dysbiosis ( 10.1016/j.obmed.2020.100302 )

Reply: done as suggested.

  1. Please also add a short comment on potential ocular complications of SARS CoV-2, such as cataract progression ( https://www.psychiatria-danubina.com/UserDocsImages/pdf/dnb_vol33_noSuppl%2013/dnb_vol33_noSuppl%2013_424.pdf)

Reply: done as suggested.

  1. Highlight that acute hyperglycaemia, hypoglycaemia, as well as hyperglycaemia post‐hypoglycaemia all produce oxidative stress and are followed by enormous production of inflammatory cytokines and by enhancement of the inflammatory/infectious process
    Reply: done as suggested.
  2. Please include primary endpoints from the most important and registration clinical trials for the most important antiviral medical agents. Consider to mention potential drug-drug interactions that prescribers need to be aware of.

Reply: done as suggested.

  1. Provide safety data for the vaccines (overview of most important and rare specific AE). Also mention which vaccines have polysorbate 80 and which PEG, which is important from the alergic point of view. Comment on booster doses of individual vaccines and associated efficacy.

Reply: done as suggested.

  1. Conclusion need to be more specific and please extend it.

Reply: conclusion extended, and Table 1 lists very specific items – but we are willing to go further if needed.